# Binding to nucleosome poises human SIRT6 for histone H3 deacetylation

Ekaterina Smirnova[1,2,3,4], Emmanuelle Bignon[5], Patrick Schultz[1,2,3,4], Gabor Papai[1,2,3,4]*, Adam Ben Shem[1,2,3,4]*

[1]Department of Integrated Structural Biology, IGBMC, Institut de Génétique et de Biologie Moléculaire et Cellulaire (IGBMC), Illkirch, France; [2]Université de Strasbourg, IGBMC UMR 7104-UMR-S 1258, Illkirch, France; [3]CNRS, UMR 7104, Illkirch, France; [4]Inserm, UMR-S 1258, Illkirch, France; [5]Université de Lorraine and CNRS, UMR 7019 LPCT, Nancy, France

**\*For correspondence:**
papai@igbmc.fr (GP);
adam@igbmc.fr (ABS)

**Competing interest:** The authors declare that no competing interests exist.

**Abstract** Sirtuin 6 (SIRT6) is an $NAD^+$-dependent histone H3 deacetylase that is prominently found associated with chromatin, attenuates transcriptionally active promoters and regulates DNA repair, metabolic homeostasis and lifespan. Unlike other sirtuins, it has low affinity to free histone tails but demonstrates strong binding to nucleosomes. It is poorly understood how SIRT6 docking on nucleosomes stimulates its histone deacetylation activity. Here, we present the structure of human SIRT6 bound to a nucleosome determined by cryogenic electron microscopy. The zinc finger domain of SIRT6 associates tightly with the acidic patch of the nucleosome through multiple arginine anchors. The Rossmann fold domain binds to the terminus of the looser DNA half of the nucleosome, detaching two turns of the DNA from the histone octamer and placing the $NAD^+$ binding pocket close to the DNA exit site. This domain shows flexibility with respect to the fixed zinc finger and moves with, but also relative to, the unwrapped DNA terminus. We apply molecular dynamics simulations of the histone tails in the nucleosome to show that in this mode of interaction, the active site of SIRT6 is perfectly poised to catalyze deacetylation of the H3 histone tail and that the partial unwrapping of the DNA allows even lysines close to the H3 core to reach the enzyme.

## eLife assessment

This manuscript provides a **useful** reconstruction of the structure of the sirtuin-class histone deacetylase Sirt6 bound to a nucleosome based on cryo-EM observations, and additional characterization of the flexibility of the histone tails in the complex based on molecular dynamics simulations. While similar structures have recently been published elsewhere, this **solid** study supports the conclusions of those papers and also includes new insights into the potential dynamics of Sirt6 bound to a nucleosome, insights that help explain its substrate specificity.

## Introduction

The state of lysine acetylation in histones, as well as in several other proteins, is determined by the balancing act of acetyltransferases and deacetylases. Sirtuin is an evolutionarily conserved family of lysine deacetylases that comprises seven members in humans and plays an important role in regulating numerous physiological and pathological processes including aging, cancer, inflammation, and metabolism (*Cen et al., 2011*; *Vitiello et al., 2017*; *Wątroba et al., 2017*). Unlike other lysine deacetylases, sirtuins carry out their enzymatic reaction by transferring the acetyl group to an essential $NAD^+$ cofactor before acetylating the target protein (*Imai et al., 2000*).

Human Sirtuin 6 (SIRT6) is a multitasking enzyme implicated in diverse pathways. Through its histone H3 deacetylation at gene promoters, it is a transcription repressor attenuating inter-alia ribosome biogenesis (*Sebastián et al., 2012*), glycolysis (*Taylor et al., 2022*) and the expression of various transcription factors, including nuclear factor κB (NF-κB; *Kawahara et al., 2009*) and hypoxia-inducible factor-1α (HIF-1α; *Zhong et al., 2010*). Its H3K9 deacetylation of telomeric chromatin maintains telomere integrity (*Michishita et al., 2008*). In addition, via its second enzymatic activity, ADP-ribosylation of PARP1 (*Mao et al., 2011*) and by stabilizing DNA-PK on chromatin (*McCord et al., 2009*), SIRT6 also fosters DNA double-strand break repair.

These activities position SIRT6 as a key regulator of longevity as indeed shown by several lines of evidence. In mice, knockout of SIRT6 leads to a shortened lifespan and a premature aging-like phenotype (*Kanfi et al., 2012*). Similar observations were made in monkeys deficient for the enzyme (*Zhang et al., 2018*). Conversely, overexpression of SIRT6 extends lifespan and diminishes frailty in mice (*Roichman et al., 2021*). Furthermore, a study with several rodent species found a positive correlation between SIRT6 ability to promote DNA double-strand break repair and longevity - in other words, more active SIRT6 leads to longer lifespan (*Tian et al., 2019*).

The development of cancer can also be regulated by SIRT6. However, whereas in some malignancies SIRT6 plays a role as a tumor suppressor (*Marquardt et al., 2013*; *Min et al., 2012*; *Sebastián et al., 2012*), in others it acts as an oncogene (*Bauer et al., 2012*; *Khongkow et al., 2013*; *Liu et al., 2013*). The molecular mechanisms underlying SIRT6 effects on longevity or on the two contradictory roles in cancer development are not well understood, limiting the search for medically useful drugs that target the enzyme.

SIRT6 is a 37 KDa protein composed of two globular parts, a Rossmann fold domain that harbors the NAD$^+$ binding site and a zinc finger domain (*Pan et al., 2011*). A striking difference between SIRT6 and other members of the sirtuin family is the presence of a splayed zinc finger and the lack of a stabilizing helix bundle linking the two globular domains. This absence probably endows SIRT6 with a unique flexibility between the domains (*Figure 1—figure supplement 1*).

These structural differences are manifested in function. Unlike other sirtuins, SIRT6 has very low activity on acetylated free H3 histones or histone peptides (*Jiang et al., 2013*). This seemed to contradict findings concerning the function of SIRT6, in particular data demonstrating its activity in vivo as H3 deacetylase (*Michishita et al., 2008*) as well as the fact that SIRT6 is nuclear and found mainly associated with chromatin (*Michishita et al., 2005*). The conundrum was settled when it was revealed that nucleosomes, the in vivo cognate substrate of SIRT6, dramatically enhance the enzyme activity toward acetylated H3 histones (*Gil et al., 2013*). Further biochemical analysis showed that SIRT6 binds nucleosomes with high affinity via multiple interacting sites (*Liu et al., 2020*).

However, a mechanistic understanding of SIRT6 binding to nucleosomes and how this association promotes higher enzyme efficacy is currently lacking.

Here, we present the structure of human SIRT6 bound to the nucleosome. We identify interactions between the zinc finger domain and the acidic patch of the nucleosome as well as between the Rossmann fold and the DNA terminus, positioning the NAD$^+$ binding pocket close to the nucleosome DNA exit site and partially detaching the DNA from the histone octamer. While this manuscript was under review two studies describing the structure of SIRT6-nucleosome appeared in press (*Chio et al., 2023*; *Wang et al., 2023*). The conclusion of these papers regarding the position of SIRT6 on the nucleosome and the unwinding of DNA by the enzyme are similar to our findings. We however use in addition molecular dynamics simulations of the nucleosome to show that in this position the enzyme active site is perfectly poised to bind the H3 tail for deacetylation and that due to the unwrapping of the DNA even lysines close to the core of H3 can reach the enzyme. Furthermore, we find conformational changes, facilitated by the flexibility in SIRT6 between the zinc finger and the Rossmann fold, that could serve to fine-tune the position of the active site with respect to the H3 tail in order to trap its target acetylated residues.

## Results
### Overall structure
To investigate how nucleosomes stimulate the activity of SIRT6 we prepared nucleosome with the Widom-601 positioning sequence (*Lowary and Widom, 1998*) and reconstituted its complex with

full-length recombinant human SIRT6 that was overexpressed in and purified from *E. coli*. The complex was stabilized via mild crosslinking before its deposition on cryo-EM grids.

We obtained a single particle cryogenic electron microscopy structure showing the nucleosome at 2.9 Å overall resolution and a protein density bound to its disc-like surface (*Figure 1a* and *Figure 1—figure supplements 2 and 3*). SIRT6 could be fitted into this density with its zinc finger domain (resolved at 3.0 Å resolution), associated with the acidic patch of the nucleosome and its Rossmann fold domain (4.0–6.0 Å resolution) interacting with the DNA terminus at superhelix locations (SHL) 6 and 7 (*Figure 1b* and *Figure 1—figure supplement 4*). The active site of SIRT6 is thus positioned just above where the H3 tail enters the nucleosome disc between the two DNA gyres. The high-resolution at the nucleosome allows discerning purines from pyrimidines. We could therefore distinguish between the two pseudosymmetric orientations of the Widom-601 DNA (*Figure 1—figure supplement 5a*). We find that SIRT6 is bound predominantly at the 'right' TA-poor arm of the Widom-601 sequence where DNA interactions with the histone octamer are looser (*Ngo et al., 2015*). We detect however a small population with a second SIRT6 bound at the 'left' TA-rich tight arm (*Figure 1—figure supplement 2*).

Conformational heterogeneity analyses in cryoDRGN (*Zhong et al., 2021*) reveal two principle large movements within the structure (*Figure 2a and d* and *Figure 2—figure supplement 1*).

First, we find that where the SIRT6 Rossmann fold binds, DNA is pulled away and detached from the histone octamer (*Figure 2b*). Particles greatly vary in the extent of this shift that therefore probably represents a continuous movement of the DNA (compare *Figure 2b1* with 2b2). The Rossmann fold moves with and is attached to the DNA whereas the zinc finger domain remains fixed at the acidic patch of the nucleosome (*Figure 2c*). In the full extent of the DNA movement, two turns from SHL 6 and 7 are detached from the histone octamer. This alters the path of the terminus DNA by 37° and breaks DNA interactions with histones H2A, H2B, and H3 (*Figure 1—figure supplement 5b*). Such a large change in DNA trajectory is reminiscent of the chromatin remodelling ATPases or pioneer transcription factors binding to nucleosome but was not observed in other histone modifiers (*Dodonova et al., 2020*; *Farnung et al., 2017*). Importantly, the DNA terminus without bound SIRT6 or free nucleosomes in our dataset did not show any deviation from the canonical conformations (*Figure 1—figure supplement 5c*). In addition, our MD simulations indicate higher flexibility of the DNA termini when SIRT6 is bound to the nucleosome (*Figure 2—figure supplement 2*). All together these findings strongly suggest that the DNA movement we observed is coupled to SIRT6 interactions.

Second, in particles where the SIRT6-bound DNA remains in its canonical position, we observe shifts by up to 20 Å in the position of the Rossmann fold domain of SIRT6, partially loosening its association with the DNA (*Figure 2e and f*). We show that this shift could be involved in fine tuning the position of the active site relative to the H3 tail.

These conformational changes are made possible by a flexible hinge region between the two domains of SIRT6, which is unique to this member of the sirtuins family.

## Zinc finger domain binds the acidic patch

An important feature of the nucleosome surface is a cluster of acidic residues (E56, E61, E64, D90, E91, E92 of H2A and E102, E110 of H2B) that forms a negatively charged 'acidic patch' and takes the shape of a narrow groove. It serves as a docking site for many nucleosome-binding proteins. Four side chains of the SIRT6 zinc-finger domain extend into different sections of the acidic patch groove and interact with the negatively charged residues. **R175** packs against H2B L103 and serves as a classical 'arginine anchor motif' (*McGinty and Tan, 2016*), binding to the acidic triad of E61, D90 and E92 from H2A. SIRT6 presents two other arginines, **R172**, to contact E56 and Q24 of H2A as well as Q44 and E110 of H2B and **R178** to contact H2A E91 and E102 from H2B. Finally, **K170** interacts with E64 and N68 from H2A (*Figure 3a*).

Substituting these four positively charged residues to alanines (SIRT6-4A) significantly weakened the interaction of SIRT6 with the nucleosomes as shown by electrophoretic gel-shift assay (*Figure 3—figure supplement 1a*). To test also the activity of SIRT6-4A, we first acetylated the H3 tails of the nucleosomes with purified SAGA complex. The mutant protein showed strongly diminished activity in deacetylating H3K9Ac, further corroborating the role of the four basic residues R175, R172, R178, and K170, in binding to the nucleosome (*Figure 3—figure supplement 1b*).

All four SIRT6-positive residues are located in a loop that makes a sharp turn, due to a conserved glycine at position 173, connecting two beta strands and penetrates the acidic patch. Interestingly,

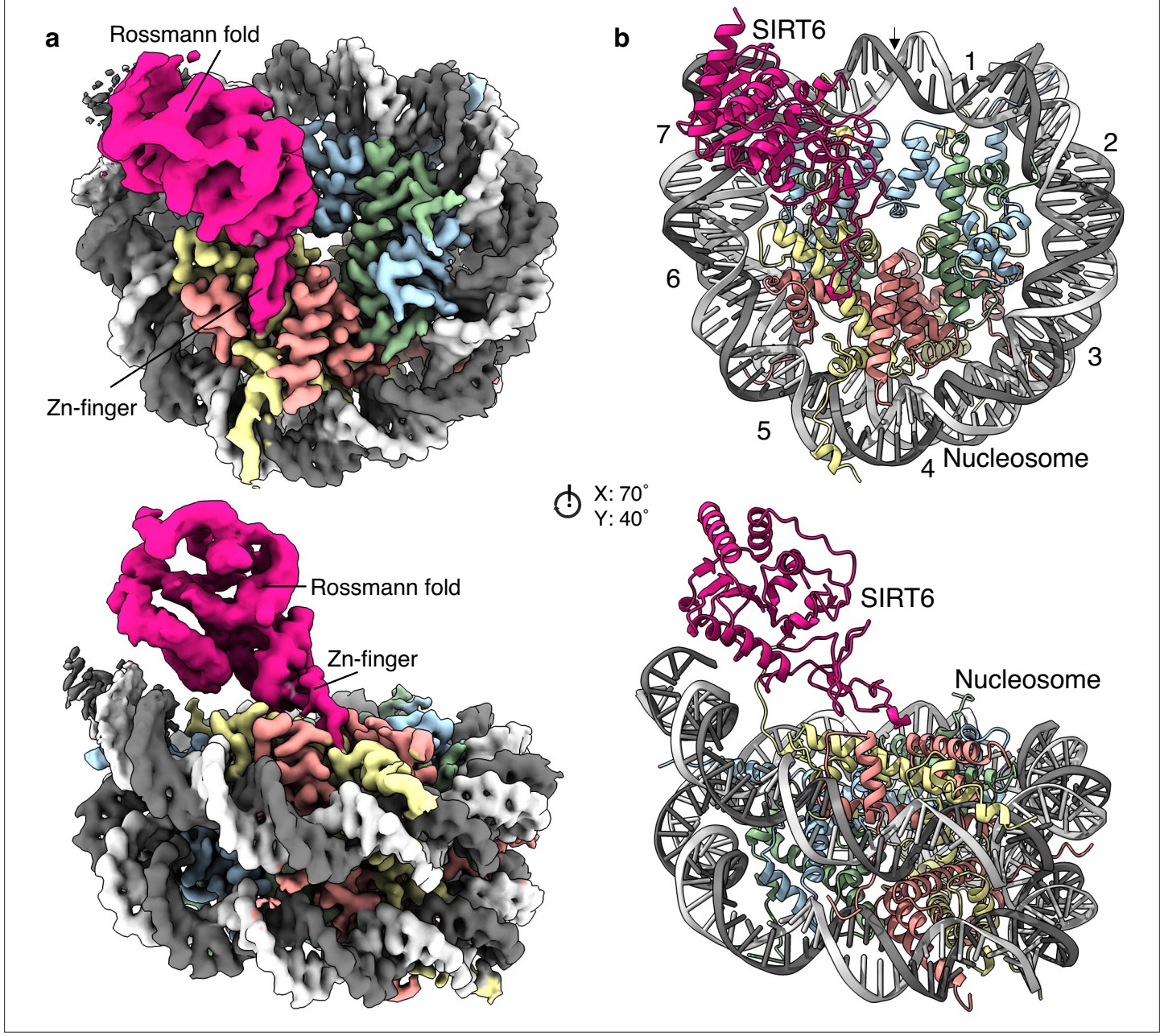

**Figure 1.** Structure of human SIRT6 in complex with the nucleosome. (**a**) Front (top) and side (bottom) views of a composite cryo-EM reconstruction of human SIRT6-nucleosome. Maps from focused refinements of SIRT6 (magenta – Rossmann fold and Zn-finger domains labelled) and the nucleosome (H2A – yellow; H2B – orange; H3 – blue; H4 – green; DNA – light and dim grey). (**b**) Corresponding views of the atomic model of the complex.

The online version of this article includes the following figure supplement(s) for figure 1:

**Figure supplement 1.** SIRT6 lacks the helix bundle between the Rossmann fold and the Zinc-finger domains.

**Figure supplement 2.** Cryo-EM data analysis strategy for SIRT6-nucleosome – dataset 1.

**Figure supplement 3.** Cryo-EM data analysis strategy for SIRT6-nucleosome – dataset 2.

**Figure supplement 4.** Representative regions illustrating the quality of the cryo-EM map of SIRT6 bound to nucleosome.

**Figure supplement 5.** SIRT6 binds to and displaces the "looser" DNA terminus.

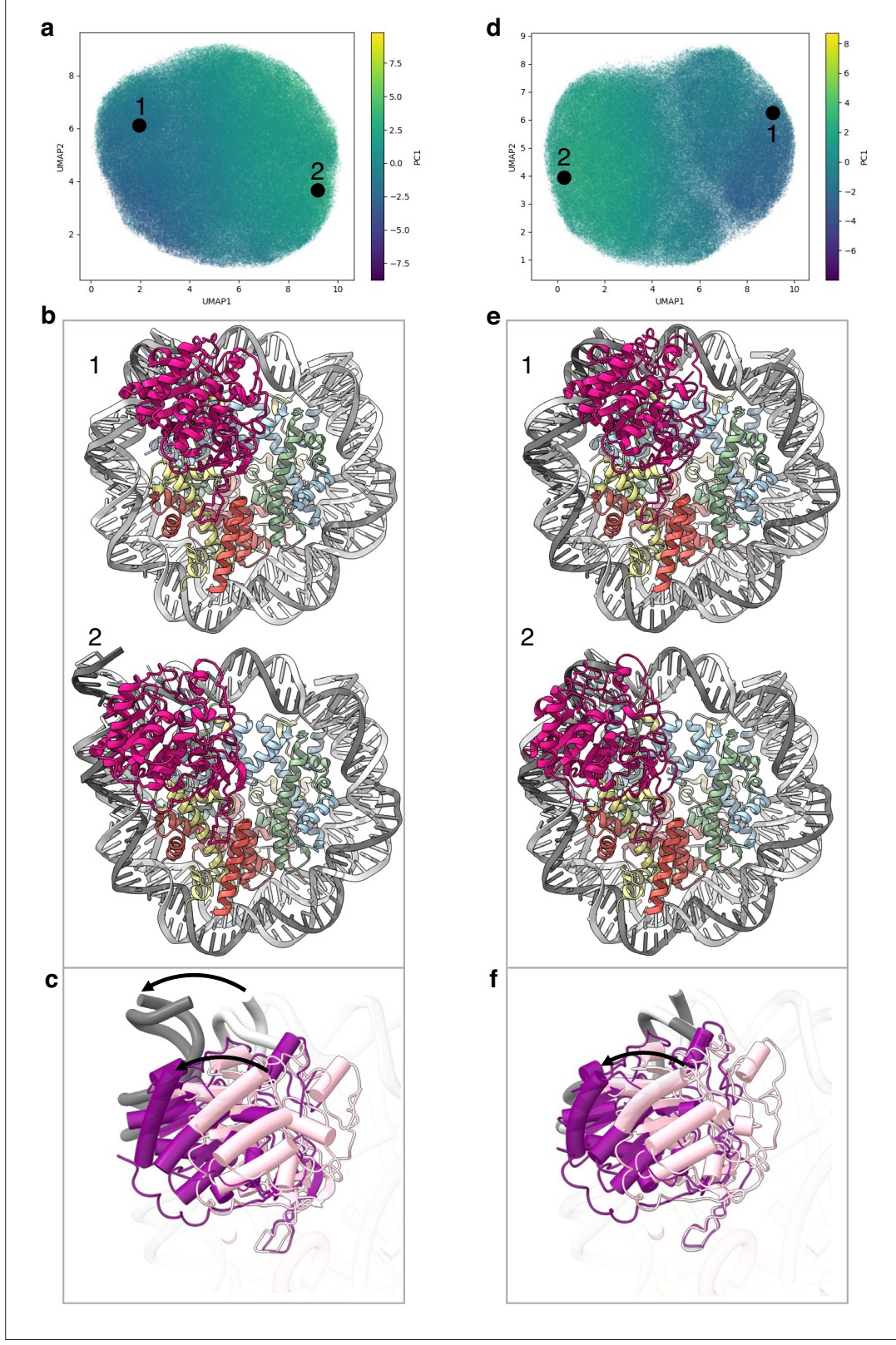

**Figure 2.** Flexibility of the SIRT6 Rossmann fold. (**a**) UMAP projection of the latent embeddings of a subset of SIRT6-nucleosome particles representing the concerted movement of the Rossmann fold and the DNA terminus. (**b**) Structural representation of the two endpoints of the latent embeddings shown in panel a. (**c**) Overlay of the DNA ends and the SIRT6 Rossmann fold of two endpoints shown in panel b. Arrows show their displacements

*Figure 2 continued on next page*

*Figure 2 continued*

between the two endpoints. (**d**) UMAP projection of the latent embeddings of another subset of SIRT6-nucleosome particles representing the movement of the Rossmann fold with respect to the nucleosomal DNA. (**e**) Structural representation of the two endpoints of the latent embeddings shown in panel b. (**f**) Overlay of the DNA ends and the SIRT6 Rossmann fold of two endpoints shown in panel e. Arrow show the displacement of the Rossmann fold between the two endpoints.

The online version of this article includes the following figure supplement(s) for figure 2:

**Figure supplement 1.** Schematic representation of CryoDRGN analyses.

**Figure supplement 2.** SIRT6 binding renders DNA termini more flexible.

this loop appears to be flexible in crystal structures of SIRT6 but rendered stable by its association with the acidic patch. Binding to different sections of the acidic patch maintains a strong hold of the zinc finger at the surface of the nucleosome and orients the Rossmann fold domain toward the DNA terminus. This strong hold is maintained throughout the movements of the Rossmann fold domain (*Figure 2*).

It is important to note that the four positive residues of SIRT6 that bind the nucleosome acidic patch are conserved among vertebrates but in contrast are not present in other human sirtuins (*Figure 3b and c*). This probably reflects the fact that SIRT6 is the only sirtuin predominantly associated with chromatin.

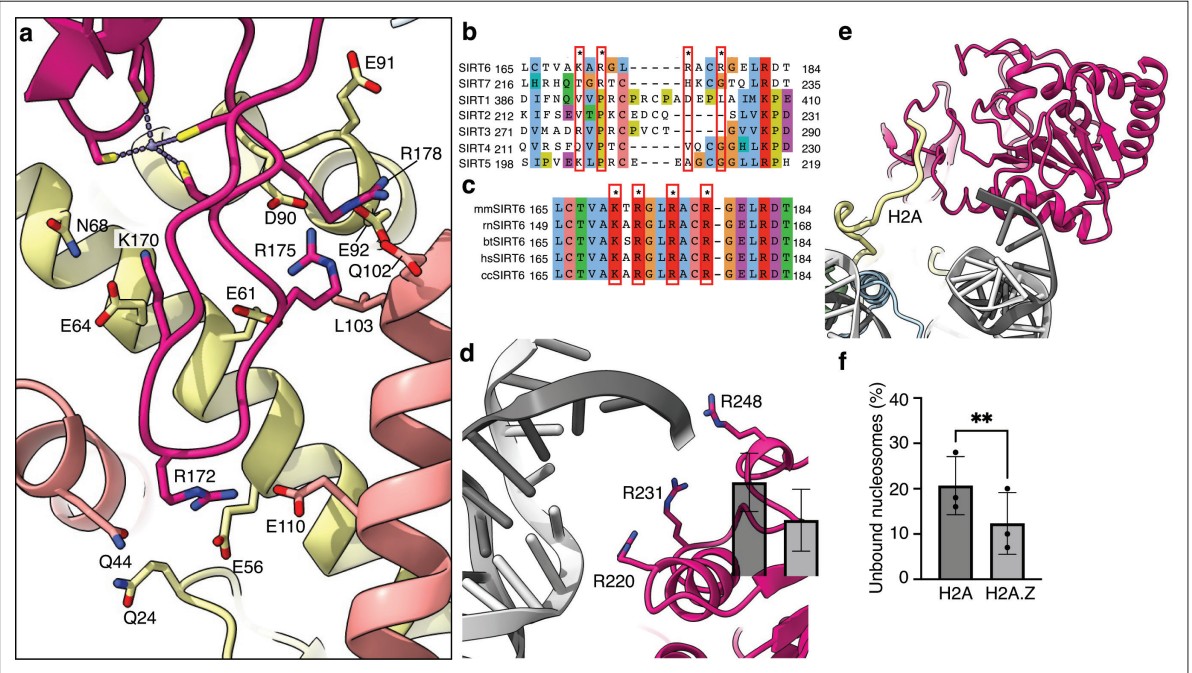

**Figure 3.** Binding of SIRT6 to the nucleosome. (**a**) Close-up view on zinc-finger interactions with the acidic patch. Color code as in *Figure 1*. (**b**) Protein sequence alignment of human sirtuins. Red boxes and asterisks depict the residues of SIRT6 interacting with the acidic patch of the nucleosome. (**c**) Protein sequence alignment of SIRT6 from different species highlighting the same amino acids as in panel b. Organisms: mm *Mus musculus*, rn *Rattus norvegicus*, bt *Bos taurus*, hs *Homo sapiens* and cc *Castor canadensis*. (**d**) Depiction of the three arginines of SIRT6 (magenta) interacting with the DNA (grey). (**e**) H2A c-terminal tail (yellow) interacts with SIRT6(magenta). (**f**) SIRT6 binding to H2A or H2AZ containing nucleosomes. Bars show the fraction of residual nucleosomes that did not shift with bound SIRT6 in an electron-mobility shift assay (*Figure 3—figure supplement 3*). Bars represent mean ± SD of three biological replicates (shown as dots). ** indicates a statistically significant difference between the fraction of residual H2A and H2A.Z containing nucleosomes (p=0.0016 in paired t-test).

The online version of this article includes the following figure supplement(s) for figure 3:

**Figure supplement 1.** WT and mutant SIRT6 - interactions with nucleosome and deacetylation activity.

**Figure supplement 2.** SIRT6 rearrangement upon nucleosome binding.

**Figure supplement 3.** Electrophoretic mobility shift assay comparing SIRT6 binding to H2A.

## Rossmann fold domain binds nucleosome DNA exit site

Due to its flexibility with respect to the fixed zinc finger domain, the resolution of the Rossmann fold domain is not sufficiently high to distinguish most individual side chains. However, secondary structure elements, mostly helices, are clearly visible and guided the precise fitting of the crystal structure of this domain.

We find that three highly conserved arginines from the Rossmann fold domain (R220, R231, R248) bind the phosphate backbone at SHL 6 and 7 of the nucleosomal DNA (*Figure 3d*). These interactions are mostly maintained throughout the movement of the DNA terminus with the Rossmann fold domain bound. However, by comparison to dedicated DNA binding proteins such as transcription factors or TBP, the number of observed DNA-protein interactions is low in SIRT6. A biochemical study suggests that the intrinsically disordered C-terminal domain of SIRT6 is necessary for establishing tight interaction of SIRT6 with the nucleosome (*Liu et al., 2020*) via its ability to bind DNA. Our maps do show an additional isolated protein density that we could not assign with confidence to SIRT6 but associates with the DNA terminus *Figure 3—figure supplement 2a*. We propose however that the three arginines are required to correctly position the Rossmann fold, and hence the catalytic site, with respect to the H3 tail. Indeed, applying SAGA-acetylated nucleosomes as substrates, we find that a SIRT6 mutant, where the three arginines are substituted to alanines (SIRT6-3A), has severely diminished H3K9Ac deacetylase activity (*Figure 3—figure supplement 1b*). These findings confirm the importance of the three lysines R220, R231, and R248 to the enzymatic function.

Unlike other sirtuins, SIRT6 binds $NAD^+$ in the absence of an acetylated substrate (*Imai et al., 2000*). This is attributed to the fact that crystal structures of SIRT6 describe a rigid helix (α3) where other sirtuins contain a flexible cofactor binding loop (*Pan et al., 2011*). In our map, we do not observe helix α3 (*Figure 3—figure supplement 2b*) and we speculate that in the context of the nucleosome, this helix assumes a flexible loop conformation that like in other sirtuins will bind the cofactor only after stabilization by the acetyl lysine substrate.

In addition, we find that the N-terminal part of the Rossmann fold domain associates with the C-terminal tail of histone H2A at residue K119 (*Figure 3e*). This observation suggests that monoubiquitination of H2A at this position, a major histone modification in mammalian cells, could interfere with the binding of SIRT6 or regulate its activity. Furthermore, we used electrophoretic mobility shift assay to follow the binding of SIRT6 to nucleosomes harboring the histone variant H2A.Z which deviates considerably from the canonical H2A in its C-terminal tail but not in other residues that contact SIRT6. Surprisingly, we find that binding of SIRT6 to nucleosomes is mildly (but statistically significant) stronger when H2A.Z is incorporated instead of H2A (*Figure 3f* and *Figure 3—figure supplement 3*). This finding could be related to SIRT6's role in repressing gene expression because H2A.Z is enriched in nucleosomes that flank the transcription start site.

## SIRT6 is poised to deacetylate the H3 tail

The classical model of the nucleosome posits an elongated and disordered histone H3 N-terminal tail that freely roams and extends away from the core. However, recent NMR studies and molecular dynamics predictions show that the tail actually collapses onto the nucleosome core, driven by robust interactions with DNA (*Ikebe et al., 2016*; *Stützer et al., 2016*). The tail adopts multiple structurally heterogeneous, but energetically similar, states. It dynamically transitions between these conformations that all interact with the last superhelix location of nucleosomal DNA (*Morrison et al., 2018*). We used a published set of 15 predicted H3-tail conformations resulting from end-states of molecular dynamics simulations of the whole nucleosome (*Morrison et al., 2018*), to analyze whether the position of SIRT6 on the nucleosome serves the enzyme's role in trapping residues of histone H3 tail. The nucleosome from each molecular dynamics simulation was superposed on our model of SIRT6 bound to a nucleosome with the DNA terminus at the canonical position. To represent the full range of movement of the Rossmann fold domain relative to the DNA, we used two endpoints of this movement namely fully bound to DNA and partially dissociated from it (*Figure 4a and b*, respectively). We then examined the position of the SIRT6 active site with respect to the predicted paths of the histone H3 tail.

We find that in several H3 conformations, the tail is located in close proximity (<15 Å) to the SIRT6 active site. In some of these cases, H3K9 or H3K18, the main target residues of SIRT6 (*Wang et al., 2016*), are practically situated in the active site. Furthermore, when analyzing the two endpoints of the

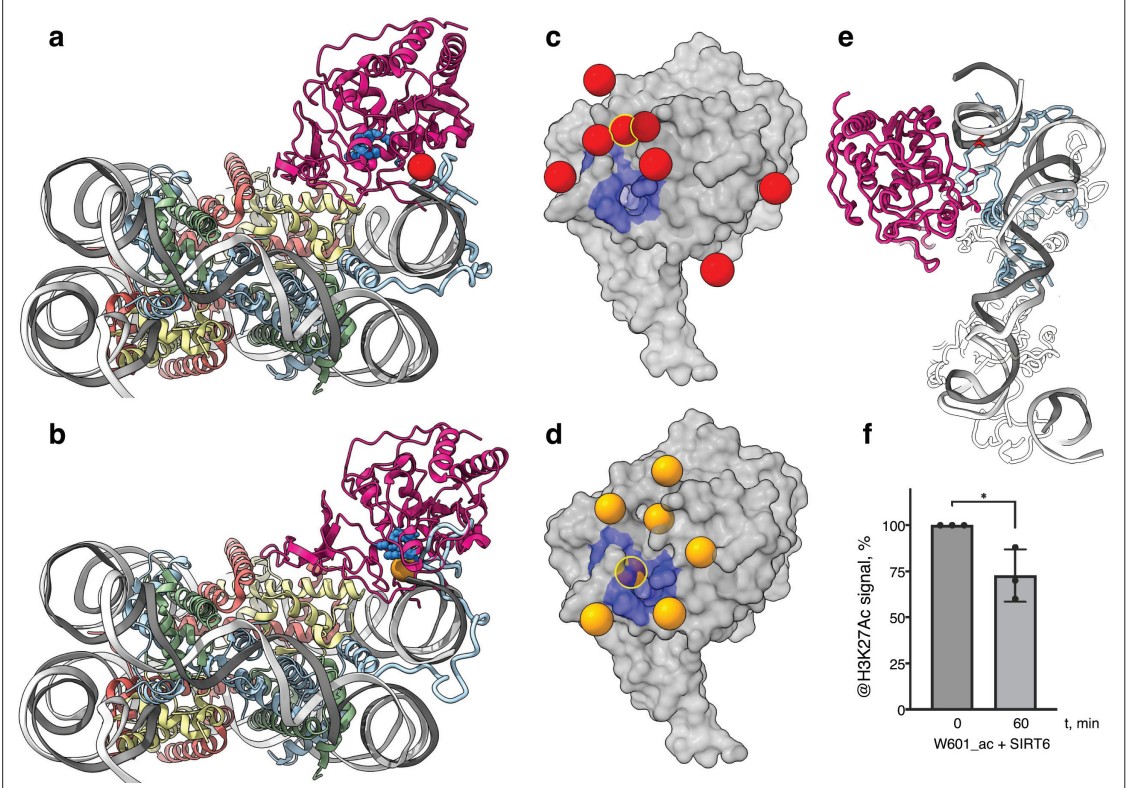

**Figure 4.** SIRT6 poised to deacetylate lysine residues of H3. (**a**) Side View of SIRT6 bound to the nucleosome with histone H3 K9 residue (red sphere) closest to the SIRT6 active site (blue spheres) from a set of molecular dynamics simulations. (**b**) Side View of SIRT6 bound to the nucleosome with histone H3 K18 residue (orange sphere) closest to the SIRT6 active site from a set of molecular dynamics simulations. (**c**) All H3K9 positions (red spheres) in close proximity (<15 Å) to the SIRT6 active site (blue) taken from a set of 15 molecular dynamics simulations and depicted on the surface view of SIRT6 bound to nucleosome. (**d**) All H3K18 positions (orange spheres) in close proximity (<15 Å) to the SIRT6 active site (blue) taken from a set of 15 molecular dynamics simulations and depicted on the surface view of SIRT6 bound to nucleosome. (**e**) Molecular dynamics simulations show that H3 c-terminal tail (blue) can protrude toward SIRT6 in a space formed between the histone octamer and the DNA. H3K27 is shown in red. (**f**) Quantification analysis of H3K27ac bands intensities in deacetylation assay. Bars show percentage of signal detected in western blot run with anti-H3K27ac antibodies. Bars represent mean ± SD of three biological replicates (shown as dots). * indicates a statistically significant difference between the 0 min (control) and 60 min (SIRT6 treatment) fraction of acetylated H3K27 (p=0.0396 in one-way paired t-test).

Rossmann fold domain shift with respect to the DNA, we observe that in some H3 conformations, this shift draws the active site closer to the target residues (*Figure 4c and d*). We conclude that the high-affinity binding of SIRT6 to the nucleosome poises the active site to trapping its main target residues. In addition, the flexibility between the Rossmann fold and Zn-finger domains serves as a fine-tuning mechanism that allows for short-range scanning of the tail in order to locate the acetylated residues. We note also that except H3 no other histone is present at the vicinity of the SIRT6 active site. This gives rise to selectivity of SIRT6 toward the H3 tail.

To shed light on possible benefits offered by SIRT6-mediated DNA unwrapping, we conducted molecular dynamic simulations of a nucleosome with bound SIRT6. Strikingly, new H3 tail conformations appear in the simulations that are absent from a control run with nucleosome alone. These pathways have the H3 tail protruding out of the histone core through the space between the DNA and the octamer that is formed via partial unwrapping of the DNA by SIRT6. Importantly, in these additional conformations H3 tail does not need to circle around the DNA to reach SIRT6 and therefore lysines that are closer to the core of H3 (e.g., K27) could also be accessible to the enzyme (*Figure 4e*). To lend support to this claim, we tested whether SIRT6 can deacetylate residue H3K27 that was first acetylated by SAGA (*Figure 4f* and *Figure 3—figure supplement 1c*). We find that indeed SIRT6 could efficiently deacetylate H3K27Ac, although at a somewhat slower rate than H3K9Ac. We conclude that partial DNA unwrapping by SIRT6 allows H3-tail conformations that make lysines that are close to the core of H3 accessible to the enzyme.

# Discussion

Most histone-modifying enzymes, including sirtuin family members such as SIRT1, show significant activity on isolated target peptides and form multiple interactions with the residue to be modified as well as with flanking amino acids to provide selectivity (*Zhao et al., 2013*). It is therefore striking that the measured turnover rate for SIRT6 on isolated H3K9 peptides is 1000 slower than for other sirtuins (*Jiang et al., 2013*). Moreover, unlike SIRT1, the activity of SIRT6 is dramatically stimulated by binding to its cognate nucleosome substrate (*Gil et al., 2013*).

We solved the structure of SIRT6 bound to a nucleosome and used heterogeneity analysis to uncover the main conformational changes in the complex. Examining these findings in the context of molecular dynamics studies of the nucleosome that produce putative paths of histone tails, shed light on the unique properties of SIRT6.

We find that multiple interactions of SIRT6 with the nucleosome position its active site at the putative path of relatively long-lived conformations of H3 so that residues K9 and K18 of the tail spend significant time very close to, or even at, the enzyme active site. In addition, SIRT6 moves with respect to the nucleosomal DNA allowing it to fine-tune the position of the active site to trap the target residue. Hence it is the binding of SIRT6 to the nucleosome and an intrinsic scanning capability that account for its much higher enzymatic efficiency on nucleosomes.

Importantly, the position of SIRT6 also ensures high selectivity because the space surrounding the active site is devoid of any other tails apart from H3.

The recognition mode of nucleosomal H3 employed by SIRT6 is reminiscent of that used by the PRC1 complex in the specific ubiquitylation of H2A Lys119 (*McGinty et al., 2014*). Like SIRT6, PRC1 requires a nucleosome substrate, showing no activity on an H2A-H2B dimer, and multiple interactions with the nucleosome position an active site in close proximity to its target residue. However, the flexibility and scanning capacity we observed for SIRT6 is lacking in PRC1 perhaps echoing the fact that its target residue is rather stable in the nucleosome compared to the H3 tail.

It is interesting to note that the flexibility of SIRT6 that is at the root of its scanning mechanism was suggested to be, at least in part, responsible for its nearly undetected activity on isolated peptides (*Pan et al., 2011*). It is only when it is constrained to the proximity of the target residues, through binding to the nucleosome, that this capacity facilitates the function of the enzyme.

A second major conformational change observed in our 3D reconstructions is the detachment of the DNA terminus, with a bound Rossmann fold domain, from the octamer. Our molecular dynamics simulations suggest that the space formed between DNA and octamer allows for additional conformations of the H3 tail that protrudes toward SIRT6. These new pathways do not require circling around the DNA gyre and therefore lysines that are closer to the core of H3 (e.g. K27) can reach in this way the active site of SIRT6. We note that K56 which lies at the core of H3 within a stable helix was also reported to be a target for SIRT6. It is therefore possible that the enzyme employs additional recognition modes of its targets.

Because the position of the SIRT6 active site seems to be well suited for modifying the H3 tail, it stands to reason that other H3 modifying enzymes, such as the GCN5 HAT module of SAGA and ATAC, would place their active site in a position similar to the one of SIRT6 and possibly also displace the DNA from the histone octamer. Gcn5 deviates from SIRT6 in having significant affinity for the H3 tail and therefore might pose a less stringent requirement for precise positioning of the active site or for specific conformations of the H3 tail. Thus, we speculate the Gcn5 recognition mode will be similar to SIRT6 but will also include some sequence specific binding.

# Materials and methods

**Key resources table**

| Reagent type (species) or resource | Designation | Source or reference | Identifiers | Additional information |
|---|---|---|---|---|
| Strain, strain background (*Escherichia coli*) | BL21-RIL | N/A | | |
| Recombinant DNA reagent | SIRT6 (plasmid) | AddGene | #28271 | SIRT6 (PDB: 3PKJ) in pET28a-LIC |

*Continued on next page*

*Continued*

| Reagent type (species) or resource | Designation | Source or reference | Identifiers | Additional information |
|---|---|---|---|---|
| Recombinant DNA reagent | Mutant SIRT6 (DNA fragment) | SynBio | | K170A, R172A, R175A, R178A, R220A, R231A and R248A |
| Recombinant DNA reagent | Widom601 DNA (plasmid) | **Dyer et al., 2004** | | Gift from Dr. K. Mohideen-Abdul. |
| Recombinant DNA reagent | *X. laevis* H2A, H2B, H3, H4 histones (plasmids) | **Luger et al., 1999**; **Dyer et al., 2004** | | Gift from Dr. K. Mohideen-Abdul. |
| Recombinant DNA reagent | Human H2A.Z.1 (plasmid) | AddGene | #42629 | H2A.Z.1 in pET28a |
| Antibody | H3K27ac (rabbit monoclonal) | CellSignaling | #8173 S | Diluted: 1:1000 |
| Antibody | H3K9ac (rabbit polyclonal) | Abcam | Ab4441 | Diluted: 1:2500 |
| Antibody | H3 (mouse monoclonal) | CellSignaling | #14269 S | Diluted: 1:1000 |
| Software, algorithm | SerialEM | | v4.1 | Automated data collection |
| Software, algorithm | EPU | Thermo Fisher | v3.4 | Automated data collection |
| Software, algorithm | Warp | Warp | v1.0.9 | Micrograph preprocessing |
| Software, algorithm | cryoSPARC | cryoSPARC | v4 | Image analysis |
| Software, algorithm | RELION | RELION | v3.1 | Image analysis |
| Software, algorithm | crYOLO | crYOLO | v1.9.6 | Particle picking |
| Software, algorithm | cryoDRGN | cryoDRGN | v1.1 | Image analysis – structural flexibility analysis |
| Software, algorithm | DeepEMhancer | DeepEMhancer | v0.8 | Map sharpening |
| Software, algorithm | UCSF Chimera/ChimeraX | UCSF ChimeraX | v1.16/1.6 | Rigid body fitting, visualisation and figures |
| Software, algorithm | Isolde | Isolde | v1.6 | Flexible fitting, model refinement |
| Software, algorithm | PHENIX | PHENIX | v1.21 | Model refinement |
| Software, algorithm | AMBER20 suit | AMBER20 | | MD simulations |

## Human SIRT6 purification

Plasmid (AddGene #28271) was transformed into *E. coli* strain BL21-RIL. Cells in 1 L TB (with 50 μg/mL Kanamycin and 25 μg/mL Chloramphenicol) grew at 37 °C degrees till $OD_{600}$ reached 0.8 when they were rapidly cooled on ice to 18 °C. Over-expression was induced by the addition 0.5 mM IPTG and cells continued to grow at 18 °C over-night. All subsequent steps were performed at 0–4 °C and solutions were always supplemented with a protease inhibitor cocktail (PMSF, Pepstatin, E-64). Cells were harvested by centrifugation at 8250 g for 20 min, suspended in buffer A200 (20 mM HepesKOH pH 7.5, 5% glycerol v/v, 2 mM beta-mercaptoethanol and 200 mM NaCl) and treated with lysosyme (1 mg/mL) for 30 min on ice. Salt concentration was increased to 0.5 M and cells were then lysed by sonication (6*10 s). Debris was pelleted by centrifugation 45,000 × g*20 min. The supernatant was incubated with 0.5 mL of Ni beads (1 mL slurry, Roche), equilibrated in buffer A500, for 1 hr. The solution was poured into a gravity column (CliniScience). Beads were then washed with 10 mL of buffer A500 with 5 mM imidazole before elution by 2 mL of buffer B300 (20 mM HepesKOH pH 7.2, 5% glycerol v/v, 2 mM beta-mercaptoethanol and 300 mM NaCl) with 150 mM imidazole. The eluted protein was diluted with two volumes of buffer D (20 mM HepesKOH pH 7.0, 5% glycerol v/v, 2 mM beta-mercaptoethanol), loaded on a 1 mL Heparin column and eluted with 0.1 M–1 M salt gradient in buffer D. Fraction were analyzed by SDS-PAGE and those containing SIRT6 were pooled and dialyzed against buffer E (20 mM HepesKOH pH 7.5, 2% glycerol v/v, 0.5 mM TCEP and 150 mM NaCl). Human SIRT6 was then concentrated to 7.8 mg/mL, flash-frozen in liquid $N_2$, and kept in aliquotes at –80 °C.

## Cloning of SIRT6 mutants

The region of AddGene plasmid #28271 between SacI and HindIII sites was synthetized by SynBio company with changes in DNA sequence leading to K170A, R172A, R175A, R178A, R220A, R231A, and R248A mutations in amino acid sequence. A NheI restriction site was also introduced between the first four and last three mutations without changes in amino acid sequence in order to facilitate cloning. A plasmid expressing SIRT6-7A that incorporates all seven mutations was created via ligation of the synthetic DNA fragment into the original expression plasmid between the SacI and HindIII sites. Fragments of the original expression plasmid corresponding to the regions between HindIII and NheI or NheI and SacI were amplified with PCR and were cloned separately through corresponding restriction sites into the plasmid expressing SIRT6-7A to yield plasmids expressing SIRT6 with the first four mutations (SIRT6-4A) or the last three mutations (SIRT6-3A), respectively.

## Octamer reconstitution and nucleosome DNA

Octamers were reconstituted from individual *Xenopus leavis* (canonical) or human (H2A.Z) histones expressed as inclusion bodies according to the standard protocol (*Dyer et al., 2004*; *Luger et al., 1999*). Widom-601 145 bp DNA was produced using a plasmid harboring 16 copies of this sequence as described by *Dyer et al., 2004*.

## Nucleosome core particle reconstitution

Nucleosomes with 145 bp Widom-601 positioning sequence were prepared according to NEB Dilution Assembly Protocol (E5350) (https://international.neb.com/protocols/2012/06/02/dilution-assembly-protocol-e5350) with some modifications as follows: 2.75 µM 145 bp Widom-601 DNA was mixed with 2.75 µM canonical or H2A.Z histone octamers in a solution containing 2 M NaCl, 1 mM EDTA, 5 mM beta-mercaptoethanol. The solution was incubated for 30 min at RT and then underwent serial dilutions down to 1.48 M, 1 M, 0.6 M, 0.25 M NaCl with buffer LowSalt (10 mM HepesKOH pH 8.0, 2.5 mM beta-mercaptoethanol). After each dilution the solution was kept at RT for 30 min. In order to reduce the final NaCl concentration, nucleosomes were concentrated in 0.5 mL 100 KDa cutoff Amicon up to 100 µL, then diluted five times with buffer LowSalt. This step was repeated one more time. Finally, nucleosomes were concentrated to 3–4 µM and analyzed in a 5% native 0.2 x TBE polyacrylamide gel to ascertain the quality of the sample and absence of free DNA.

## Nucleosome acetylation

Canonical Widom-601 nucleosomes (final concentration 1.8 µM) were mixed with the holo-SAGA complex (final 18 nM) purified from *Pichia pastoris* cells according to *Papai et al., 2020* in 20 mM HepesKOH pH 7.5, 10% glycerol v/v, 50 mM NaCl, 2 mM $MgCl_2$, 5 mM beta-mercaptoethanol, 0.005% Tween-20, and 0.05 mM Acetyl-CoA. Reaction was incubated on ice for 16 hr. On the next day SAGA was removed via incubation of the mixture with Streptavidin Sepharose beads. In order to reduce the remaining Acetyl-CoA concentration, nucleosomes were concentrated in 0.5 mL 100 KDa cutoff Amicon up to 100 µL, then diluted five times with buffer 10 mM HepesKOH pH 8.0, 2.5 mM beta-mercaptoethanol. This step was repeated four more times. Finally, the nucleosomes were concentrated to 3.5 µM.

## Binding and gel-shift assays

For nucleosome-binding assays, 100 nM of recombinant nucleosomes (harboring either H2A or H2A.Z histone) were incubated in binding buffer (10 mM HepesKOH pH 7.5, 50 mM NaCl, 0.5 mM TCEP, 6% glycerol v/v) with SIRT6 at 1:2 molar ratio. The reactions were performed on ice for 1 hr in LoBind tubes (Eppendorf) to minimize protein loss. The reactions were then resolved in a 5% 0.2 x TBE gel. The gel was stained with Ethidium Bromide. Bands were quantified with ImageJ.

## Deacetylation assay

Non-acetylated or SAGA-acetylated canonical Widom-601 nucleosomes were mixed with SIRT6 (wt, SIRT6-4A or SIRT6-3A) at 100 nM and 40 nM, respectively, in 50 mM HepesKOH pH 7.5, 1 mM DTT, 0.2 mg/mL BSA, 1 mM $NAD^+$. Reactions were set on ice and then incubated at 37 °C for 1 or 2 hr. Reactions were quenched by boiling with 5 x Laemmli buffer. The proteins were separated on 15% SDS-PAAG and acetylation was analyzed by Western blot using the following antibodies: for

H3K27Ac CellSignaling #8173 S diluted 1:1000; for H3K9Ac Abcam Ab4441 1:2500; for H3 CellSignaling #14269 S 1:1000.

## Grid preparation

SIRT6 was first diluted in buffer (20 mM HepesKOH pH 7.5, 150 mM NaCl, 0.5 mM TCEP, 2% Trehalose). Nucleosome and SIRT6 were then mixed at a 1:2 molar ratio. Final conditions included 1.7 μM Nucleosome and 3.4 μM SIRT6 in buffer (20 mM HepesKOH pH 7.5, 18 mM NaCl, 0.5 mM TCEP). After 1 hr on ice cross-linking agent glutaraldehyde and the detergent dodecyl-maltoside were added at a final concentration of 0.05% v/v and 0.0125% w/v, respectively. The sample was incubated with the cross-linker for an additional 30–60 min. Two nm carbon foil was floated on 3.5/1 300 mesh Quantifoil grids. Three μL of sample was deposited onto such grids, blotted 5 s with blot force 5 after 60 s incubation in a Vitrobot IV (Thermo Fisher) at 6 °C using 100% humidity and flash-frozen in liquid ethane.

## cryo-EM data acquisition

Two different datasets were collected. For the first dataset, images were acquired on a Cs-corrected Titan Krios G1 (Thermo Fisher) microscope operating at 300 kV in nanoprobe mode using serialEM for automated data collection. Movie frames were recorded on a Gatan K3 direct electron detector after a Quantum LS 967 energy filter using a 20 e$^-$V slit width in zero-loss mode. Images were acquired hardware-binned at a nominal magnification of 81,000, which yielded a pixel size of 0.862 Å. Forty movie frames were recorded at a dose of 1.12 and 1.30 $e^-$ per Å$^2$ per frame. The second image dataset was acquired on a Titan Krios G4 microscope operating at 300 kV in nanoprobe mode using EPU for automated data collection. Movie frames were recorded on a Flacon 4i direct electron detector after a Selectris X energy filter using a 10 e$^-$V slit width in zero-loss mode. Images were acquired at nominal magnification of 270,000, which yielded a pixel size of 0.458 Å.

## Image analysis

For the first dataset movie frames were aligned, dose-weighted in Warp (*Tegunov and Cramer, 2019*) together with CTF estimation and particle picking. For the second dataset frame alignment, dose-weighting and CTF estimation was done in cryoSPARC (*Punjani et al., 2017*), picking was done with crYOLO (*Wagner et al., 2019*). These datasets were analyzed in cryoSPARC and RELION 3.1 (*Zivanov et al., 2018*) according to standard protocols. Briefly, images were subjected to reference-free 2D classification in cryoSPARC to remove images with contaminations. Ab-initio reconstruction was performed on the selected images followed by heterogenous and homogenous refinements in cryoSPARC. Structural variability and flexibility were explored with Relion 3.1 using 3D classification without alignment with different T values and cryoDRGN (*Zhong et al., 2021*) according to standard protocols. Sharpening of the maps was done with DeepEmhancer (*Sanchez-Garcia et al., 2021*).

## Model building and refinement

A composite map was created from the part corresponding to the SIRT6 Rossmann-fold from the structure of the first dataset and the SIRT6 Zn-finger plus nucleosome from the structure of the second dataset. Crystal structure of Widom-601 sequence containing nucleosome (PDB: 3LZ0) and human SIRT6 (PDB: 5X16) were rigid-body fitted into this composite cryo-EM structure in Chimera (*Pettersen et al., 2004*). Flexible-fittings were performed with Isolde (*Croll, 2018*) in ChimeraX (*Goddard et al., 2018*) into the cryoDRGN structures. All models were refined in real-space with PHENIX (*Liebschner et al., 2019*) and Isolde.

## Depiction of SIRT6 – nucleosome structures

The following maps are depicted in the figures:

*Figure 1*: EMD-16845; *Figure 2*: b1-cryoDRGN run1 PC1 volume 0, b2-volume 9; e1-cryoDRGN run2 PC1 volume 0, e2-volume 9; *Figure 3*: a,d-EMD-16845, e-EMD-18497; *Figure 4*: a- cryoDRGN run2 PC1 volume 0, b-volume 9; *Figure 1—figure supplement 4*: a,b-EMD-16842, c-EMD-16843, d-EMD-18497; *Figure 1—figure supplement 5*: a-EMD-16842, c- cryoDRGN run1 PC1 volume 0 and nucleosome void of SIRT6 (*Figure 1—figure supplement 2*), d- nucleosome void of SIRT6; *Figure 3—figure supplement 2*: EMD-18497.

## Molecular dynamics simulations

All molecular dynamics (MD) simulations were performed with the AMBER20 suit of programs, using the ff14SB (*Maier et al., 2015*) force field with bsc1 DNA corrections (*Ivani et al., 2016*) and CUFIX ions corrections (*Yoo and Aksimentiev, 2018*) that improve the description of disordered region conformations. Parameters for the zinc finger were taken from *Macchiagodena et al., 2020*.

The histone tails were added to the cryo-EM structure based on the work described by *Armeev et al., 2021*, and protonation states of the protein residues were checked with the propka3 software (*Olsson et al., 2011*). The system was soaked in a truncated octrahedral TIP3P water box of 20 Å buffer and 0.15 mM of NaCl ions were added to simulate physiological conditions, resulting in a total of 330,000 atoms.

The minimization of the energy was carried out in five steps, each one alternated with a 20ps NVT equilibration step at 100 K to ensure a very smooth optimization of the starting geometry. Each minimization run included 5000 steps in steepest descent and 5000 steps in conjugate gradient, with decreasing restraints on the protein and nucleic atoms from 20kcal/mol to 5kcal/mol that were kept in the associated equilibration step. The last minimization step was released of any constraints and followed by a final NVT equilibration run at 300 K. The time step was then switched from 2 to 4fs as allowed by the use of the SHAKE bond length constraints (*Kräutler et al., 2001*) and the Hydrogen Mass Repartioning algorithm (*Hopkins et al., 2015*). The latter was applied on the topology using the parmed tool of Amber20. A 40ns equilibration run in NTP was then carried out, followed by a 5µs production. Coordinates were outputted every 1ns during the production run. The temperature was kept constant using the Langevin thermostat with a 2ps$^{-1}$ collision frequency, and the Berendsen barostat was used to maintain the pressure at 1 atm in the NTP runs. Electrostatic interactions were treated using the Particle Mesh Ewald approach (*Darden et al., 1993*) with an 8 Å cut-off. Three replicates were performed starting with different velocities at the unconstrained 300 K NVT equilibration step, resulting in a total of 15µs of sampling.

A cluster analysis was performed on the MD ensembles based on the DNA and protein RMSD. Conformations were extracted every 5 frames of the trajectories to avoid memory overload and were sorted using the hierarchical agglomerative and average linkage algorithms. Ten clusters were generated by replicate (*Supplementary file 1b*).

The DNA flexibility analysis was performed on each MD ensemble by using a Principal Component Analysis-based machine learning script quantifying the contribution of each residue to the overall dynamics of the system. Based on the Scikit-learn58 library (*Fleetwood et al., 2020*), it calculates a covariance matrix from the internal coordinates of the MD ensemble. In this matrix, the eigenmodes showing the highest amplitude are the most implicated in the main fluctuations of the system, and the contribution of each residue is normalized based on these values.

The control system was generated by removing SIRT6 from the initial structure and was submitted to the same protocol.

## Acknowledgements

We acknowledge support from the Institut National de la Santé et de la Recherche Médicale (INSERM), the Centre National pour la Recherche Scientifique (CNRS), the Ligue contre le Cancer. We acknowledge the use of resources of the French Infrastructure for Integrated Structural Biology FRISBI ANR-10-INBS-05 and of Instruct-ERIC. EB thanks GENCI (HPC resources of IDRIS, allocation A0120713412) for computational resources.

## Additional information

### Funding

| Funder | Grant reference number | Author |
| --- | --- | --- |
| Ligue Contre le Cancer | | Ekaterina Smirnova<br>Patrick Schultz<br>Gabor Papai<br>Adam Ben Shem |

| Funder | Grant reference number | Author |
|---|---|---|
| French Infrastructure for Integrated Structural Biology | ANR-10-INBS-05 | Ekaterina Smirnova<br>Patrick Schultz<br>Gabor Papai<br>Adam Ben Shem |
| GENCI | A0120713412 | Emmanuelle Bignon |

The funders had no role in study design, data collection and interpretation, or the decision to submit the work for publication.

### Author contributions

Ekaterina Smirnova, Conceptualization, Data curation, Formal analysis, Validation, Investigation, Writing - original draft, Project administration, Writing - review and editing; Emmanuelle Bignon, Validation, Investigation, Visualization, Writing - original draft, Writing - review and editing; Patrick Schultz, Conceptualization, Funding acquisition, Writing - original draft; Gabor Papai, Conceptualization, Data curation, Formal analysis, Supervision, Validation, Investigation, Visualization, Methodology, Writing - original draft, Project administration, Writing - review and editing; Adam Ben Shem, Conceptualization, Data curation, Formal analysis, Supervision, Validation, Investigation, Methodology, Writing - original draft, Project administration, Writing - review and editing

### Author ORCIDs

Emmanuelle Bignon  http://orcid.org/0000-0001-9475-5049
Patrick Schultz  http://orcid.org/0000-0002-7310-6186
Gabor Papai  http://orcid.org/0000-0002-2779-8679

Reviewer #1 (Public Review): https://doi.org/10.7554/eLife.87989.5.sa1
Reviewer #3 (Public Review): https://doi.org/10.7554/eLife.87989.5.sa2
Author Response https://doi.org/10.7554/eLife.87989.5.sa3

---

## Additional files

### Supplementary files

• Supplementary file 1. Supplementary information for cryo-EM and molecular dynamics symulations. (a) Cryo-EM data collection, refinement and validation statistics. (b) Frequency of conformational clusters for each MD ensemble. Clusters featuring the H3 tail protruding between the DNA and the octamer are marked by a star.

• MDAR checklist

### Data availability

The cryo-EM maps have been deposited in the Electron Microscopy Data Bank (EMDB) under accession codes EMD-16842 (High-resolution on the nucleosome), EMD-16843 (SIRT6-nucleosome refined for SIRT6), EMD-18497 (Tracing the H2A tail) and EMD-16845 (composite map). CryoDRGN maps are available at https://github.com/papaig/SIRT6-cryoDRGN. The model coordinates for SIRT6-nucleosome were deposited in the Protein Data Bank (PDB) under the accession code 8OF4. All MD input files are publicly available on Github: https://github.com/emmanuellebignon/NCP-SIRT6 copy archived at *Bignon, 2024*.

The following datasets were generated:

| Author(s) | Year | Dataset title | Dataset URL | Database and Identifier |
|---|---|---|---|---|
| Smirnova E, Bignon E, Schultz P, Papai G, Ben-Shem A | 2023 | SIRT6 bound nucleosome | https://www.ebi.ac.uk/emdb/EMD-16842 | Electron Microscopy Data Bank, EMD-16842 |

*Continued on next page*

*Continued*

| Author(s) | Year | Dataset title | Dataset URL | Database and Identifier |
|---|---|---|---|---|
| Smirnova E, Bignon E, Schultz P, Papai G, Ben-Shem A | 2023 | Nucleosome bound human SIRT6 | https://www.ebi.ac.uk/emdb/EMD-16843 | Electron Microscopy Data Bank, EMD-16843 |
| Smirnova E, Bignon E, Schultz P, Papai G, Ben-Shem A | 2023 | Human Sirtuin 6 in complex with nucleosome - structure showing H2A tail path | https://www.ebi.ac.uk/emdb/search/EMD-18497 | Electron Microscopy Data Bank, EMD-18497 |
| Smirnova E, Bignon E, Schultz P, Papai G, Ben-Shem A | 2023 | Nucleosome Bound human SIRT6 (composite structure) | https://www.ebi.ac.uk/emdb/search/EMD-16845 | Electron Microscopy Data Bank, EMD-16845 |
| Smirnova E, Bignon E, Schultz P, Papai G, Ben-Shem A | 2023 | Nucleosome Bound human SIRT6 (Composite) | https://www.rcsb.org/structure/8OF4 | RCSB Protein Data Bank, 8OF4 |
| Papai G | 2024 | SIRT6-cryoDRGN | https://github.com/papaig/SIRT6-cryoDRGN | GitHub, SIRT6-cryoDRGN |

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
