## [Editor Report · eLife assessment]

This manuscript provides a **useful** reconstruction of the structure of the sirtuin-class histone deacetylase Sirt6 bound to a nucleosome based on cryo-EM observations, and additional characterization of the flexibility of the histone tails in the complex based on molecular dynamics simulations. While similar structures have recently been published elsewhere, this **solid** study supports the conclusions of those papers and also includes new insights into the potential dynamics of Sirt6 bound to a nucleosome, insights that help explain its substrate specificity.

---

## [Referee Report · Reviewer #1 (Public Review)]

Smirnova et al. present a cryo-EM structure of a nucleosome-SIRT6 complex to understand how the histone deacetylase SIRT6 deacetylates the N-terminal tail of histone H3. The authors obtained the structure at sub-4 Å resolution and can visualize how interactions between the nucleosome and SIRT6 position SIRT6 to allow for H3 tail deacetylation. Through additional conformational analysis of their cryo-EM data, they reveal that SIRT6 positioning is flexible on the nucleosome surface, and this could accommodate the targeting of certain H3 tail residues. This work is significant as it represents the visualization of a histone deacetylase on its native nucleosomal target and reveals how substrate specificity is achieved. Importantly, it should be noted that recently two additional structures of the nucleosome-SIRT6 complex were already published. Therefore, Smirnova et al. confirm and complement these previous findings. Additionally, Smirnova et al. expand our understanding of the structural flexibility of SIRT6 on the nucleosome and clarify that SIRT6 also shows histone deacetylase activity on H3K27Ac.

---

## [Referee Report · Reviewer #3 (Public Review)]

Smirnova et al. present a cryo-EM structure of human SIRT6 bound to a nucleosome as well as the results from molecular dynamics simulations. The results show that the combined conformational flexibilities of SIRT6 and the N-terminal tail of histone H3 limit the residues with access to the active site, partially explaining the substrate specificity of this sirtuin-class histone deacetylase. Two other groups have recently published cryo-EM structures of SIRT6:nucleosome complexes; this manuscript confirms and complements these previous findings, with the addition of some novel insights into the role of structural flexibility in substrate selection.

---

## [Author Response]

The following is the authors’ response to the previous reviews.

We thank the reviewers for their remarks. Please find our detailed answers below.

1. The authors' continued refusal to acknowledge the other reports before the final sentence of the Discussion, which has been pointed out in two previous rounds of review as a major flaw, detracts from the manuscript significantly.

We now acknowledge and discuss the other SIRT6-nucleosome reports in the introduction as requested by the reviewer.

1. While some of the grammatical errors in previous versions have been corrected, many remain, especially in the Methods section

We corrected the remaining grammatical errors.

1. Multiple statements of fact not supported by data shown in this work continue to lack appropriate references.

We added references where facts were not supported by our data.